# A Systematic Comparative Study on the Physicochemical Properties, Volatile Compounds, and Biological Activity of Typical Fermented Soy Foods

**DOI:** 10.3390/foods13030415

**Published:** 2024-01-27

**Authors:** Qingyan Guo, Jiabao Peng, Yujie He

**Affiliations:** 1Food Microbiology Key Laboratory of Sichuan Province, School of Food and Bioengineering, Xihua University, Chengdu 610039, China; 19150356036@163.com (J.P.); 0120210019@mail.xhu.edu.cn (Y.H.); 2Chongqing Key Laboratory of Speciality Food Co-Built by Sichuan and Chongqing, Chengdu 610039, China

**Keywords:** fermented soy foods, E-nose, flavor profile, antioxidant, antimicrobial, correlation analysis

## Abstract

Fermented soy foods can effectively improve the unpleasant odor of soybean and reduce its anti-nutritional factors while forming aromatic and bioactive compounds. However, a differential analysis of characteristic flavor and function among different fermented soy foods has yet to be conducted. In this study, a systematic comparison of different fermented soy foods was performed using E-nose, HS-SMPE-GC×GC-MS, bioactivity validation, and correlation analysis. The results showed that soy sauce and natto flavor profiles significantly differed from other products. Esters and alcohols were the main volatile substances in furu, broad bean paste, douchi, *doujiang*, and soy sauce, while pyrazine substances were mainly present in natto. Phenylacetaldehyde contributed to the sweet aroma of furu, while 1-octene-3-ol played a crucial role in the flavor formation of broad bean paste. 2,3-Butanediol and ethyl phenylacetate contributed fruity and honey-like aromas to douchi, *doujiang*, and soy sauce, respectively, while benzaldehyde played a vital role in the flavor synthesis of douchi. All six fermented soy foods demonstrated favorable antioxidative and antibacterial activities, although their efficacy varied significantly. This study lays the foundation for elucidating the mechanisms of flavor and functionality formation in fermented soy foods, which will help in the targeted development and optimization of these products.

## 1. Introduction

As the inclination towards promoting holistic wellbeing advances, an escalating populace is delving into plant-based diets as a viable nutritional substitute [1]. Among these substitutes, soybeans are a significant candidate. Soybeans contain abundant lipids (19.9%), carbohydrates (30%), and plant-based protein (36.5%), and their proportion is similar to that of meat products, making soybeans one of the most suitable meat substitutes [2]. In addition, soybeans contain various functional ingredients, including vitamins, isoflavones, bioactive peptides, saponins, and plant sterols, which make their use as dietary supplements common [3,4]. However, the consumption of raw soybeans is restricted due to undesirable odors and anti-nutritional components.

In numerous regions, including Asia and Europe, fermentation is a widely utilized technique of food preservation [5]. Moreover, fermentation is a primary process for producing food from soybeans. Traditional fermented soy foods (FSFs) encompass a wide array of items such as furu, broad bean paste (BBP), douchi, *doujiang*, soy sauce, and natto. Throughout the process of soybean fermentation, microorganisms engage in robust metabolic activity to break down macromolecular organics, leading to the synthesis of a variety of compounds, including peptides, amino acids, and fatty acids [6] while concurrently producing flavor and biologically active compounds [7]. According to research findings, fermentation effectively removes protease inhibitors that hinder nutrient absorption [8]. The concentration of vitamin K2 and vitamin B12 demonstrate a notable increase, while the assimilation of copper and iron is significantly enhanced [9]. These findings suggest a noteworthy improvement in the bioavailability of said nutrients. Consequently, FSFs are currently gaining global attention and demand.

In essence, FSFs are a food with a distinct taste obtained through microbial fermentation of soybeans. The fermentation process involves many biochemical reactions that result in changes to the physicochemical properties while enriching their sensory characteristics and imparting a unique flavor. Jung et al. identified *Bacillus* and other genera in *Dajiang*, and found that these aerial microbes significantly contributed to the development of the unique flavors of *Dajiang* [10]. During pre-fermentation, mold secretes proteases that degrade soybean protein into amino acids and peptides, generating sour, sweet, and salty flavors. The biochemical processes that involve amino acids, encompassing transamination, dehydrogenation, decarboxylation, and esterification, generate compounds with gustatory properties, namely, aldehydes, acids, alcohols, and esters [11]. Furthermore, it can be observed that lactic acid bacteria and mold secrete a significant quantity of lipase, leading to the liberation of fatty acids. Following their interaction with ethanol, these fatty acids produce ethyl-flavoring substances. This chemical process directly influences the formation of the flavor profile of FSFs [12]. However, the flavor of FSFs varies significantly due to differences in fermentation technology, microbial species, fermentation environment, and the quality and properties of the soybean varieties used for fermentation [13]. In particular, most soybeans (more than 95%) are genetically modified, and the number of varieties has drastically decreased because large seed and chemical companies dominate the seed market. Currently, there needs to be a more systematic comparison and summary of characteristic flavors among different FSFs, the lack of which dramatically limits the development and optimization of differentiation of FSFs. Flavor is the main characteristic of fermented products. Hence, an in-depth exploration of the correlation between flavor and different FSFs is an effective way to achieve precise quality control of fermented bean products.

On the other hand, microbial fermentation is accompanied by the degradation of organic matter, which significantly improves biological utilization while effectively promoting the production of bioactive ingredients, endowing FSFs with multiple health benefits. Multiple studies have shown that FSFs can significantly alleviate hyperlipidemia and lower the likelihood of developing atherosclerosis and cardiovascular pathologies [14]. The *Bacillus subtilis* natto derivatives exhibit a significant inhibitory effect towards *Enterococcus faecalis*, which is recognized as a major etiology of nosocomial infections [15]. Yang et al. examined *Lactococcus acidophilus* fermented soymilk and found that the FSFs showed significantly higher antioxidant activity than non-fermented products [16]. Relevant research on 20 different types of furu demonstrated that furu possesses a unique taste and aroma and has strong anti-mutagenic and antioxidant properties [17]. The peptides RGLSK and TPPCPQ, isolated from *Pixian* broad bean paste, exhibited remarkable antioxidant and ACE inhibition activities, further underscoring the importance of FSFs as a reservoir of biologically active agents [18]. However, a comparative analysis of the differences in the biological functions targeted toward different FSFs has yet to be conducted.

Therefore, in this research we aimed to comprehensively compare six typical FSFs, namely, furu, BBP, douchi, *doujiang*, soy sauce, and natto, concerning their physicochemical properties, volatile compounds, and biological activities. The E-nose and HS-SPME-GC × GC-MS were employed to accurately capture the flavor differences among different FSFs. Further correlation analysis was performed to elucidate the effects of physico-chemical parameters and flavor substances on the characteristic flavor of different FSFs. Furthermore, the combination of in vitro antioxidant and antimicrobial activity tests elucidated the functional differences among these products. This study will lay the foundation for differentiating the spectral maps of different FSFs and provide a basis for targeted development and deep utilization based on the functional differences between different products.

## 2. Materials and Methods

### 2.1. Materials

All FSFs were bought on the market. A total of six FSFs representative of the general population were chosen. Three different brands of each FSF were randomly selected; the information is shown in Table 1. For the same FSF, brands from different regions were selected as much as possible in order to show their differences. However, as BBP is a specialty fermented soybean product from Sichuan, the three selected brands are the most common as representatives for daily consumption. Specifically, furu is fermented tofu, also called fermented bean curd, white bean-curd cheese, or tofu cheese. In English it is sometimes referred to as “soy cheese”. Broad bean paste (BBP) is a fermented bean paste typically made from ground soybeans which is indigenous to the cuisines of East, South, and Southeast Asia. Conversely, douchi is also known as fermented black soybeans or Chinese fermented black beans. *Doujiang* is made by fermenting soybeans after they are fried and ground up. As we all know, soy sauce is a traditional liquid condiment. Natto is a traditional Japanese food made from whole soybeans fermented with *Bacillus subtilis* var. natto.

### 2.2. Sample Preparation

The sample processing conditions referred to the Chinese national standard with minor modifications [19]. Samples (5.0 g) were homogenized after being combined with 50 mL of distilled water (T18, IKA instruments Co., Guangzhou, China). Centrifugation (5804R, Eppendorf AG, Hamburg, Germany) at 10,000 rpm for 15 min at 4 °C was followed by a 30 min ultrasonic extraction procedure (SB-5200DTN, Scientz Biotechnology Co., Ningbo, China) at 35 °C. For physicochemical examination, the produced supernatant was filtered via a 0.45 μm filter. Samples for biological activity assays were freeze-dried and dissolved in distilled water prior to analysis, while samples for other analyses were maintained at −80 °C.

### 2.3. Determination of Basic Physicochemical Characteristics

pH value: A pH meter was used to measure the pH (METTLER TOLEDO Instruments Co., Shanghai, China).

Moisture content: Fresh FSFs were dried to a consistent weight in a constant-temperature oven set at 65 °C and the starting weight ratios were compared to the final ratios. The following equation was used to determine the moisture content:(1)$$M_{c}=\frac{m_{w}-m_{d}}{m_{w}}\times 100\%,$$
where: 

$M_{c}$—Moisture content (%);$m_{w}$—Wet weight (g);$m_{d}$—Dry weight (g).

Salinity: First, 5 mL of the sample was pipetted into a 150 mL conical flask, followed by additions of 100 mL of distilled water and 1 mL of potassium chromate solution (50.0 g/L). The mixture was mixed well and titrated with silver nitrate standard solution (0.1 mol/L) until an initial orange color was obtained. Another 100 mL of distilled water was taken for the reagent blank test. Following the recording of the titration volume, the salinity value was determined by Chinese national standard [20].

Total acid/amino nitrogen: A 5 mL sample solution was mixed with 30 mL of distilled water and mixed with a magnetic stirrer. Sodium hydroxide standard solution (0.05 mol/L) was added and titrated to pH 8.2 and the volume of standard solution consumed was recorded. Formaldehyde solution (36%) was added to the mixture and again titrated to pH 9.2 and the volume of standard solution consumed was recorded. Separately, pH 8.2 was established in 35 mL of distilled water with sodium hydroxide standard solution, followed by adding 10 mL of formaldehyde solution and adjusting to pH 9.2 using sodium hydroxide standard solution. The total acid and amino nitrogen value calculation followed Chinese national standard [20].

Total sugars: First, 9.5 mL of distilled water and 500 μL of sample solution were combined in a 15 mL centrifuge tube and mixed. In addition, 1 mL of sample dilution was mixed with 1 mL of phenol solution and 5 mL of concentrated sulfuric acid was added promptly. After standing for 10 min, the mixture was mixed well using a vortex shaker and placed in a 30 °C water bath for 20 min. A standard curve for the glucose solution was used to determine the total sugar content.

Reducing sugars: A test tube containing 1 mL of sample and 1 mL of DNS solution was filled, mixed for 5 min at 100 °C, and immediately cooled in ice water for 5 min, then the absorbance at 550 nm was measured. The reducing sugar content was then determined using the glucose standard curve.

### 2.4. E-Nose Analysis

An E-nose analysis was conducted utilizing a portable electronic nose 3 (PEN3) system (Ensoul Technology LTD., Beijing, China) fitted with ten metal oxide semiconductor (MOS) gas sensors (Table 2). To achieve headspace equilibrium, a 2.0 g sample was positioned in a glass vial with a capacity of 25 mL and allowed to incubate at 25 °C for 30 min. The headspace gas was then continuously injected into the sensor array for an 80 s measurement at a 400 mL/min rate. Each experiment was followed by a 120 s clean air purge of the system. For further study, the average response value of each sensor for a 69 s period was chosen [21].

### 2.5. Determination of Volatile Compounds Using HS-SPME-GC × GC-MS

Samples were analyzed for their volatile compounds using HS-SPME combined with GC × GC-MS. The SPME and instrument parameters were established based on previous methods [22] with minor modifications.

A 15 mL SPME vial (Supelco, Inc., Bellefonte, PA, USA) was filled with 2.0 g of sodium chloride, 2.0 g of the sample, and 8 mL of distilled water. Samples were sonicated for 30 min in a water bath heated to 30 °C, then equilibrated at 60 °C for 25 min to evaporate the chemicals. A 75 μm CAR/PDMS-coated SPME fiber probe (Supelco, Inc., Bellefonte, PA, USA) was used to absorb volatile components for 25 min in an SPME vial. The fiber was then desorbed for 1 min by inserting it into the GC injection port.

The volatile flavor compounds were detected by GC × GC-MS (QP 2010 PLUS, Shimadzu, Japan). The 30 m × 0.25 mm × 0.25 μm DB wax quartz capillary column that made up the 1D column was used. The 2D column used a solid-state thermal modulator HV (c720-21005), and its dimensions were 1.2 m × 0.18 mm × 0.18 μm for the DB-17MS capillary column. Helium gas with a purity level of >99.9999% was used in the analysis as the carrier for the splitless injection. The temperature was raised to 230 °C during the study at a rate of 5 °C/min after being initially set at 40 °C for 2 min. The temperature was then maintained at 230 °C for 5 min, and a 4 s modulation period was used to synchronize the 2D analysis time with the 1D column.

By comparing the retention index and mass spectra of reference standards that were obtained from the mass spectral library of the National Institute of Standards and Technology (NIST), it was possible to confirm the identity of the volatile compounds. By normalizing peak areas, the quantification of the relative content was established.

### 2.6. Determination of In Vitro Antioxidant Activity

#### 2.6.1. DPPH Radical Scavenging Activity

The Wei et al.method was used to measure DPPH radical scavenging activity with a modest modification [23]. Three test tubes were filled with an equal volume of sample solution (2 mg/mL) and 0.1 mmol/L DPPH radical solution, an equal volume of sample solution and 95% ethanol, and an equal volume of 95% ethanol and DPPH radical solution, respectively. After 30 min of dark storage at room temperature (25 °C), the mixture’s absorbance at 517 nm was calculated (FlexA-200, ALLSHENG Instrument Co., Hangzhou, China). The following equation was used to calculate the scavenging activity:(2)$$A_{d}=\frac{1-\left(A_{i}-A_{j}\right)}{A_{0}}\times 100,$$
where: 

$A_{d}$—DPPH radical scavenging activity (%);$A_{i}$—The absorbance of the test group;$A_{j}$—The absorbance of the control group;$A_{0}$—The absorbance of the blank group.

#### 2.6.2. ABTS^+^ Radical Scavenging Activity

A modified version of the procedure created by Silva et al.was used to evaluate the ability to scavenge 2,2’-azinobis-(3-ethylbenzthiazoline-6-sulphonate) (ABTS^+^) radicals [24]. The ABTS^+^ working solution was prepared by mixing 7 mmol/L ABTS and 2.45 mmol/L potassium persulfate at a 3:1 ratio and storing it at 4 °C in darkness for 12–16 h. Phosphate buffer (5 mmol/L, pH 7.4) was used to adjust the absorbance of the ABTS^+^ working solution at 734 nm to 0.70 ± 0.02. The solution and sample solution (2 mg/mL) were combined in a 10:1 ratio and the mixture was allowed to react at room temperature (25 °C) for 6 min. At 734 nm, the absorbance was detected (FlexA-200, ALLSHENG Instrument Co., Hangzhou, China). For the blank group, distilled water was utilized in place of the sample solution. The following equation was used to calculate the scavenging activity:(3)$$A_{a}=\left(1-\frac{A_{s}}{A_{b}}\right)\times 100,$$
where: 

$A_{a}$—ABTS^+^ radical scavenging activity (%);$A_{s}$—The absorbance of the test group;$A_{b}$—The absorbance of the blank group.

#### 2.6.3. Hydroxyl Radical Scavenging Activity

A method similar to that described by Agrawal et al.was used to measure the hydroxyl radical scavenging activity with a few minor modifications [25]. In a ratio of 1:1:1, 10 mM hydrogen peroxide, 9 mmol/L ferrous sulfate, and 10 mg/mL sample solution were combined; 10 min of incubation at 37 °C was required. Then, an equal volume of 9 mmol/L salicylic acid was added and the mixture was allowed to react for 30 min at room temperature (25 °C). The absorption was detected at 510 nm (FlexA-200, ALLSHENG Instrument Co., Hangzhou, China). For the control group, distilled water was utilized in place of the sample solution. The scavenging activity was calculated using the equation below:(4)$$A_{h}=\left(1-\frac{A_{i}}{A_{0}}\right)\times 100,$$
where: 

$A_{h}$—Hydroxyl radical scavenging activity (%);$A_{i}$—The absorbance of the test group;$A_{0}$—The absorbance of the control group.

#### 2.6.4. Superoxide Anion Radical Scavenging Activity

The technique used to measure the superoxide anion radical scavenging activity was as described by Xia et al.with a minor modification [26]. A solution of 50 mmol/L Tris-HCl buffer (pH 8.0) and 6 mmol/L pyrogallic acid was pre-warmed at 25 °C for 25 min. The preheated Tris-HCl buffer, pyrogallic acid solution, and sample solution (2 mg/mL) were mixed in a ratio of 5:1:1. The reaction was stopped by adding concentrated hydrochloric acid, and the absorbance was detected at 420 nm (FlexA-200, ALLSHENG Instrument Co., Hangzhou, China). For the control group, distilled water was used in place of the pyrogallic acid, and for the blank group, distilled water was used in place of the sample solution. The following equation was used to calculate the scavenging activity:(5)$$A_{s}=\frac{A_{0}-A_{1}+A_{2}}{A_{0}}\times 100,$$
where: 

$A_{s}$—Superoxide anion radical scavenging activity (%);$A_{0}$—The absorbance of the blank group;$A_{1}$—The absorbance of the control group;$A_{2}$—The absorbance of the test group.

### 2.7. Determination of In Vitro Antimicrobial Activity

With a slight modification from Aween et al. [27], the inhibitory activity of FSFs against *Escherichia coli*, *Staphylococcus aureus*, *Listeria monocytogenes*, and *Listeria seeligeri* was evaluated in vitro. Pathogenic bacteria adjusted to 10^6^ CFU/mL in nutritional broth were added to the sample solution (2 mg/mL). Using an enzyme linked immunosorbent assay (ELISA) microplate reader (FlexA-200, ALLSHENG Instrument Co., Hangzhou, China), absorbance was measured at 600 nm after the mixture had been incubated at 37 °C for 24 h. The following equation was used to calculate the percentage of growth inhibition:(6)$$A=\frac{\left(A_{n2}-A_{n1}\right)-\left(A_{s2}-A_{s1}\right)}{A_{n1}}\times 100,$$
where: 

A—Inhibition (%);$A_{n1}$—The absorbance of control group at 0 h;$A_{n2}$—The absorbance of control group at 24 h;$A_{s1}$—The absorbance of test group at 0 h;$A_{s2}$—The absorbance of test group at 24 h;

### 2.8. Statistical Analysis

Each experiment had three consecutive runs (*n* = 3). The SPSS 25.0 program (SPSS Inc., Chicago, IL, USA) and Duncan’s Multiple range test were utilized for the one-way ANOVA. OriginPro 2022 software (OriginLab Corporation, Northampton, MA, USA) was used to process the research data graphically. Differences that were statistically significant were denoted by *p*-values of 0.05.

## 3. Results and Discussion

### 3.1. Differences in the Physicochemical Properties of Typical Fermented Soy Foods

The evaluation of physicochemical parameters is of paramount importance for FSFs. Table 3 displays the fundamental physicochemical characteristics of six typical FSFs. The outcomes show that the moisture content of the samples differed substantially. Among these products, douchi had the lowest moisture content, while the moisture content of the other products ranged from 41.400% to 70.179%. The moisture content is crucial as a medium for microbial growth and metabolism in food fermentation. It affects microbial activity and the progression of the Maillard reaction [28]. Nonetheless, it is not the primary indicator of FSF quality, and is rather a reflection of the internal microbial metabolism.

The pH value and total acid level of fermented foods are crucial quality indicators that affect the final quality and maturity. Microorganisms hydrolyze sugars, lipids, and proteins in raw materials during fermentation, creating active compounds such as peptides, amino acids, short-chain fatty acids, and organic acids. As a result, the total acid level of the fermentation system increases and the pH value decreases [29]. The pH values ranged from 5.390 to 7.930 in six typical FSFs. BBP and douchi had lower pH values, while natto had the highest, and there were significant differences between different brands. Correspondingly, the total acid level in douchi was the highest, while the total acid level in furu and natto was lower. Research showed that organic acids were the carbon source for the growth of microorganisms in the late fermentation stage. Not only do they have a unique flavor, they serve as precursors for synthesizing other flavor compounds [30]. Therefore, it can be inferred that lactic acid bacteria, yeasts, and other microorganisms decompose glucose into a significant amount of organic acid, increasing the total acid content in the system during the fermentation of douchi.

The three brands of BBP exhibited significantly higher salinity levels than other products, with values of 16.146 g/100 g, 17.862 g/100 g, and 16.692 g/100 g, respectively. In contrast, the salinity of the three brands of natto ranged from 0.234 to 0.468 g/100 g, while the remaining four products had salinity levels between 7.254 and 12.324 g/100 g. The quantity of salt supplied during the fermentation process may be responsible for these outcomes. Given the preservative properties of salt, it likely played a role in selecting beneficial microorganisms during fermentation [31]. There were significant differences in amino nitrogen levels among the six typical FSFs. Among them, the level of amino nitrogen in BBP was the lowest, ranging from 0.214 g/100 g to 0.311 g/100 g. Analysis of the BBP manufacturing process revealed a protracted fermentation phase during which the Maillard reaction and the growth metabolism of microorganisms contributed to the drop in amino nitrogen levels [32]. On the other hand, with the enhancement of public health awareness there is increasingly more controversy surrounding high-salt foods. People believe that excessive salt intake harms human health and may lead to cardiovascular and cerebrovascular diseases. Therefore, natto, with the lowest salt content, represents a healthy food that can meet people’s low-salt requirements.

Reducing sugars serve as crucial carbon sources for microbial metabolism and development and act as Maillard reaction substrates [33]. The amount of reducing sugars in furu and douchi were significantly higher than in the others. Studies have shown that reducing sugars are derived from the hydrolysis of starch and are further consumed by microorganisms and the Maillard reaction [34]. Therefore, the lower level of reducing sugars in BBP could be attributed to their consumption by a large number of microorganisms during its post-fermentation process. In contrast, the higher levels in furu and douchi derived from the action of large amounts of amylase secreted by microorganisms. Additionally, there was consistency between the levels of total sugars and reducing sugars.

Based on the Spearman correlation analysis of different physicochemical data of FSFs, it can be concluded that the pH value has significant negative correlations with salinity (*p* < 0.001) and total acids (*p* < 0.05) (Figure 1). The utilization of peptides, amino acids, and sugars by lactic acid bacteria and yeast is known to be facilitated by high-salt environments. In contrast, the metabolism of lactic acid bacteria contributes to the accumulation of acidic components in the fermentation system, resulting in a decrease in pH value and an increase in total acid level [35], which is consistent with the findings in Figure 1. However, high salinity decreases the activity of protease and amylase, affecting the formation of flavor compounds such as peptides and free amino acids. Therefore, formation of the characteristic flavor of FSFs requires precise control of salinity. Furthermore, there was a strong positive correlation between the amino nitrogen content and total sugars (*p* < 0.05). This can be attributed to the secretion of large amounts of enzymes by microorganisms, which then work synergistically to degrade proteins, fats, and carbohydrates in the system, converting them into free amino acids and organic acids [36] and contributing to the further formation of flavor compounds. In short, reasonably controlling the salinity of FSFs could inhibit the growth of pathogenic microorganisms while ensuring the growth of beneficial microorganisms, thereby helping to form the characteristic flavor of FSFs and release bioactive substances.

### 3.2. Differences in the Flavor Properties of Typical Fermented Soy Foods

#### 3.2.1. E-Nose Analysis

The scent characteristics of several FSFs were evaluated using an E-nose (Figure 2). As shown in Figure 2a, the volatile fingerprint information of different FSFs showed similarity, though with differences in specific values. This was consistent with the results of Rasooli Sharabiani et al. [37], that is, the fingerprints of different rice varieties showed remarkable similarity, displaying the same pattern with different values. Specifically, all six typical FSFs produced almost minimal reactions to five sensors (W1C, W3C, W6S, W5C, W1W, and W3S). On the other hand, all six products showed high response values for sensors W5S, W1S, W2S, and W2W, with significant differences between products that could be used to differentiate between FSFs. Among them, furu exhibited a higher response than other products in terms of W5S, W1S, W2S, and W2W, indicating that during its fermentation process a large number of nitrogen oxide, methane, alcohols, and organic sulfur compounds were produced jointly to form the unique flavor characteristics of furu. The flavor response of soy sauce was similar to that of furu, though the response was significantly lower, with only the response of W5S being comparable. Therefore, different levels of similar flavor compound compositions provide FSFs with different flavor characteristics. In addition, the signal strength of the W5S and W1S sensors was relatively low for BBP, douchi, *doujiang*, and natto. There were still differences among the samples, indicating that these four types of FSFs all produced a small number of aromatic compounds, alkanes, and methane during the fermentation process. However, their abundance varied, giving each product a unique flavor.

Furthermore, the E-nose data from different FSFs were subjected to principal component analysis (PCA), with the outcomes displayed in Figure 2b. The total variance contribution rates for PC1 and PC2 were found to be 90.4%, suggesting that these two principal components could potentially convey relevant flavor-related information for distinguishing between different FSFs. The results indicated that furu and soy sauce were positively distributed on the PC1 axis. At the same time, the remaining four FSFs demonstrated a negative distribution on the PC1 axis, leading to significant separation. Moreover, furu was positively separated on the PC2 axis, while soy sauce mainly presented negative separation on the PC2 axis, which achieved a certain degree of separation. Based on the analysis of PC1 and PC2, the flavor similarity among the four samples of BBP, douchi, *doujiang*, and natto was relatively high, and they were distinct from furu and soy sauce.

#### 3.2.2. Volatile Compounds Analysis

The characteristic flavors of distinct FSFs result from various factors encompassing the environment, raw materials, production processes, and microorganisms involved in the production process [38]. Investigation of the volatile compounds present in different FSFs was conducted using the HS-SPME-GC × GC-MS. In order to exclude differences in results between different products of the same FSF due to differences in ingredients, processes, etc., only the volatile flavor compounds present in all three products of the same FSF were analyzed (Appendix A). The volatile flavor compounds identified in the six typical FSFs mainly included aldehydes, esters, ketones, alcohols, phenols, acids, hydrocarbons, furans, sulfides, ethers, and pyrazines, which are similar to the components found in other FSFs [39]. According to Figure 3, the most important volatile compounds in furu, BBP, douchi, *doujiang*, and soy sauce were esters and alcohols. Because of their low threshold and floral fragrances, esters and alcohols have long been used to enhance the flavor of various items [40]. Unlike other compounds, pyrazines were the most significant volatile compounds in natto, accounting for 54.20%, 38.81%, and 40.92% of the total amount in the three different brands, respectively. Pyrazines are essential compounds responsible for food flavor, possessing unique sensory characteristics and with a low flavor threshold that significantly affects food sensory perception [41]. Therefore, the high levels of pyrazine compounds in natto may be critical in producing its distinctive flavor.

Volatile compounds primarily constitute a distinct taste with a low threshold, and their relative prevalence may fail to accurately depict their actual impact on the collective aromatic properties [42]. Therefore, evaluation of the contribution of volatile compounds to flavor is commonly performed using the relative odor activity value (ROAV) [43]. A higher ROAV suggests a more significant impact on the overall flavor. A component is regarded as a major flavor compound when the ROAV is ≥1, indicating that it significantly affects and contributes to the scent of the food [44]. Using HS-SPME-GC × GC-MS, 8, 10, 10, 9, 12, and 3 volatile compounds with ROAV ≥ 1 were found in furu, BBP, douchi, *doujiang*, soy sauce, and natto, respectively (Table 4). These may be the critical aroma contributors of FSFs. The results indicate that 1-octen-3-ol, which had a mushroom and earthy odor, was formed by enzyme-catalyzed reactions of unsaturated fatty acids and had a significant part in the flavor creation of BBP, douchi, *doujiang*, and soy sauce [45]. Research has shown that the formation of 1-octen-3-ol in soy sauce is closely related to *Weissella*, *Staphylococcus*, *Bacillus*, *Leuconostoc*, *Trichosporon*, and *Pichia* [46]. 2,3-Butanediol contributes fruity and creamy flavors to douchi, *doujiang*, and soy sauce, with its flavor production associated with *Zygosaccharomyces* and *Citrobacter* [47]. Ethyl phenylacetate had rOAV values of 2.443, 22.368, and 10.460 in douchi, *doujiang*, and soy sauce, respectively. It can be formed through phenylalanine metabolism and possesses a rich and sweet honey-like aroma [48]. Phenylacetaldehyde, with its sweet honey and cocoa-like flavor, contributed significantly to the sweet aroma of furu, BBP, *doujiang*, and soy sauce. Meanwhile, benzaldehyde presented a prominent almond and sweet taste with an ROAV value of 1.088, which was crucial to the flavor formation of douchi. Moreover, even though esters and alcohols made up the majority of the flavorings in soy sauce, a diverse array of crucial flavor compounds were present. Among these, 4-ethyl-2-methoxyphenol is a widely used food additive, with a relative content of approximately 3.146% and an ROAV value of 3.989. It is thought to improve the overall flavor quality of soy sauce and contribute to the creation of its distinctive flavor [49]. The results corroborated those of Liu et al. [50]. Additionally, pyrazine compounds contributed significantly to the flavor of natto and soy sauce, with di-methylpyrazine and tri-methylpyrazine particularly crucial in developing their barbeque, nutty, and cocoa flavors.

Following the Spearman correlation coefficient, further research was conducted on the correlation between physicochemical parameters and flavor compounds as well as the correlation between flavor compounds (Figure 4). The only volatile compounds we examined were those present in all six FSFs. The correlation between pH and salinity level and flavor compounds was found to be opposite, consistent with previous results, indicating a negative correlation between pH value and salinity. Benzaldehyde, 3-(methylthio) propionaldehyde, glucobrassicin, and furfuryl alcohol were negatively correlated with pH but positively correlated with salinity, indicating that properly increasing the salinity level could help to accumulate beneficial flavors such as baked potato flavor, caramel flavor, and sweetness in FSFs. A substantial negative connection was discovered between the quantity of amino nitrogen and hexanal (*p* < 0.05). Furthermore, it was discovered that the concentration of reducing sugars had a substantial positive association (*p* < 0.05) with 3-ethyl-5-(2-ethylbutyl) octadecane and a significant negative correlation (*p* < 0.01) with furfuryl alcohol. The microbial degradation of starch resulted in the formation of a large amount of reducing sugar, which was further converted into flavor substances such as alcohols, fostering the development of distinctive flavors in FSFs.

The oxidative degradation of unsaturated fatty acids produces the secondary products of alcohols, which are then converted into esters through esterification reactions with low-saturated fatty acids [51]. Furthermore, aldehydes can be converted to acids and alcohols through oxidation and reduction, and are in turn converted into esters by further esterification [52]. As shown in Figure 4b, benzaldehyde was negatively correlated with 1-octen-3-ol and ethanol, while hexanal was negatively correlated with dimethylsilanediol, ethanol, and furfuryl alcohol, indicating that benzaldehyde and hexanal may be precursors of these alcohols. Moreover, ethyl phenylacetate was negatively correlated with dimethylsilanediol, hexaethylene glycol, octaethylene glycol, heptaethylene glycol, and 1-pentanol, while ethyl palmitate was negatively correlated with dimethylsilanediol, hexaethylene glycol, 2,3-butanediol, 1-pentanol, and furfuryl alcohol, suggesting that these alcohols may be transformed into these two esters through esterification, imparting FSFs with fruity and sweet flavors. 1-Octen-3-ol and ethanol were positively correlated with both esters, revealing that these two alcohols are not precursors and jointly contribute to the beneficial flavors of FSFs.

### 3.3. Differences in the In Vitro Biological Activity of Typical Fermented Soy Foods

#### 3.3.1. Antioxidant Properties

Due to their high concentration of physiologically active compounds, including flavonoids and phenolic compounds, soybeans are an ideal source of antioxidants [53]. According to reports the level of phenolic substances in FSFs was significantly increased compared to unfermented soybeans, and the antioxidant activity was significantly enhanced [54]. Six typical FSFs were tested for their antioxidant ability utilizing a combination of the DPPH, ABTS^+^, hydroxyl radical, and superoxide anion radical tests. All six FSFs exhibited significant differences in radical scavenging activity (Table 5). Soy sauce showed the most robust scavenging activity against DPPH radical (80.107–91.643%). The increased activity was ascribed to the high quantity of total polyphenols; this result was comparable with that reported by Aoshima and Ooshima [55]. Soy sauce, however, had the lowest level of scavenging activity (22.491–25.087%) against superoxide anion radicals among the six items. Douchi exhibited the highest superoxide anion radical scavenging activity, though it was closely matched by soy sauce for hydroxyl radical scavenging activity (40.149–52.085%). According to reported findings, the antioxidant activity of douchi extract may potentially stem from its proton-donating ability [56]. Furu and BBP exhibited similar antioxidant activity, with weaker scavenging activity against ABTS^+^ radical but higher scavenging activity against the other three radicals. Both *doujiang* and natto demonstrated significant scavenging activity against four types of radicals, qualifying them as excellent candidates for natural antioxidant screening. Based on factors such as raw materials, fermentation process, and fermenting strains, soy-based fermented foods could accumulate different antioxidant active components, leading to differences in antioxidant activity. 

#### 3.3.2. Antimicrobial Properties

Antimicrobial substances derived from fermented foods are gaining increasing attention as natural alternatives to chemical antimicrobial agents. The antibacterial activity against *Escherichia coli*, *Staphylococcus aureus*, *Listeria monocytogenes*, and *Listeria seeligeri* varied noticeably across the six common FSFs (Table 6). The inhibitory activity of the same FSFs against targeted pathogenic bacteria may vary due to differences in the structure and composition of cell walls, cell membranes, and other factors. Among them, furu, BBP, douchi, and soy sauce showed the most potent inhibitory activity against *L. monocytogenes*, while *doujiang* and natto exhibited the strongest inhibitory activity against *L. seeligeri*. Furu and soy sauce showed similar inhibitory spectra against the four pathogenic bacteria, with *L. monocytogenes* > *L. seeligeri* > *E. coli* > *S. aureus*. Overall, BBP exhibited the most vigorous inhibitory activity against *E. coli*, *L. monocytogenes*, and *L. seeligeri* among the six FSFs, while its inhibitory activity against *S. aureus* was moderate. *Doujiang* showed higher inhibition activity against *S. aureus* than the other five FSFs. However, its advantage was insignificant, and its inhibitory activity against *E. coli* was markedly lower than that of other products. Microorganisms can break down proteins, carbohydrates, and other nutrients found in soybeans to create a variety of secondary metabolites, some of which have antibacterial properties. Nevertheless, differences in raw materials, fermentation processes, and microorganisms can lead to variations in the bioactive components of different FSFs, resulting in different antibacterial effects. A peptide made up of 45 amino acid residues was extracted from natto by Kitagawa et al. [57] and exhibited antibacterial action against a range of pathogenic microorganisms. Furthermore, a lipopeptide extracted from *Chungkookjang*, a traditional Korean FSF, exhibited extensive antimicrobial properties against diverse strains of Gram-positive bacteria [58]. In addition, compared with unfermented bitter beans, fermentation has shown the ability to significantly increase low-molecular-weight peptides in bitter beans, thereby enhancing their antibacterial activity [59]. Based on the findings above, it was determined that the tested FSFs included a large number of antibacterial compounds and represent a high-quality source for screening natural antibacterial agents from plants, although there are currently few publications on antibacterial components in products such as furu and BBP.

## 4. Conclusions

In this study, we conducted a systematic analysis and comparison of six typical FSFs. The results revealed a significant negative correlation between the pH value and salinity and the total acid content of the FSFs. In contrast, a significant positive correlation was observed between the amino nitrogen and total sugars. The characteristic flavor profiles of furu and soy sauce were significantly different from those of the other products. 1-Octen-3-ol was shown to be a significant flavor contributor to BBP, douchi, *doujiang*, and soy sauce, while pyrazine compounds were discovered to be essential to the distinctive flavors of natto and soy sauce. Spearman correlation analysis indicated that benzaldehyde, 3-(methylthio)propionaldehydehe, glucobrassicin, and furfuryl alcohol were positively correlated with salinity, which contributed to the accumulation of sweet taste in FSFs. While all six types of FSFs exhibited significant antioxidant and antibacterial activities, their activities showed significant differences. These results can contribute to distinguishing the flavor characteristics of different FSFs and laying the foundation for the differentiated development of traditional FSFs. However, further exploration is required in order to elucidate the process–flavor–function network of FSFs in detail. Our results contribute to the differentiated characterization of flavor characteristics of different FSFs, and can help to elucidate the fermentation process–flavor–function system. However, as only three different products of each FSF were selected, comprehensive coverage was not achieved. Furthermore, the flavor of FSFs was directly related to the microbial community, and comprehensive construction of the fermentation process–flavor–function correlation network still requires exploration of the microbial ecosystem of FSFs to provide theoretical references for the targeted development and utilization of FSFs.

## Figures and Tables

**Figure 1 foods-13-00415-f001:**
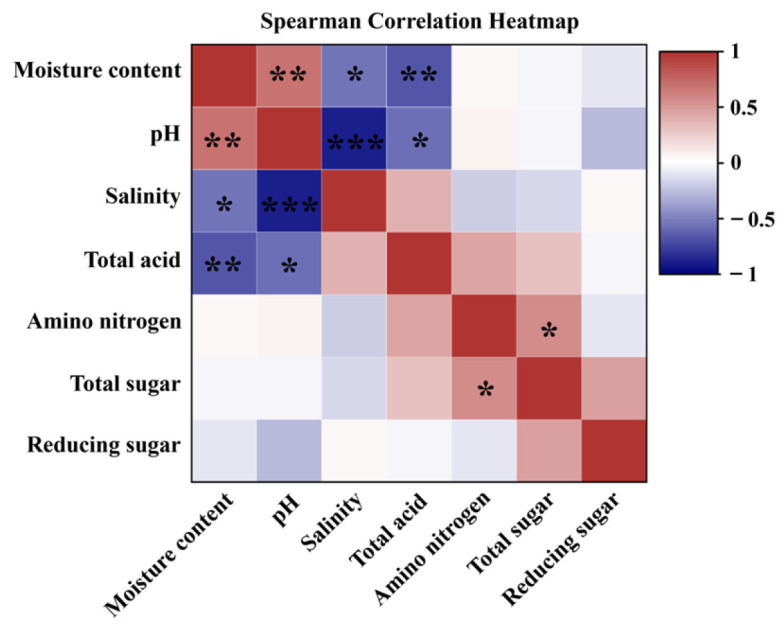
Heat map of the correlation between key physicochemical parameters of six different fermented soy foods. Notes: * represents the correlation between physicochemical parameters (***: *p* < 0.001, **: *p* < 0.01, *: *p* < 0.05).

**Figure 2 foods-13-00415-f002:**
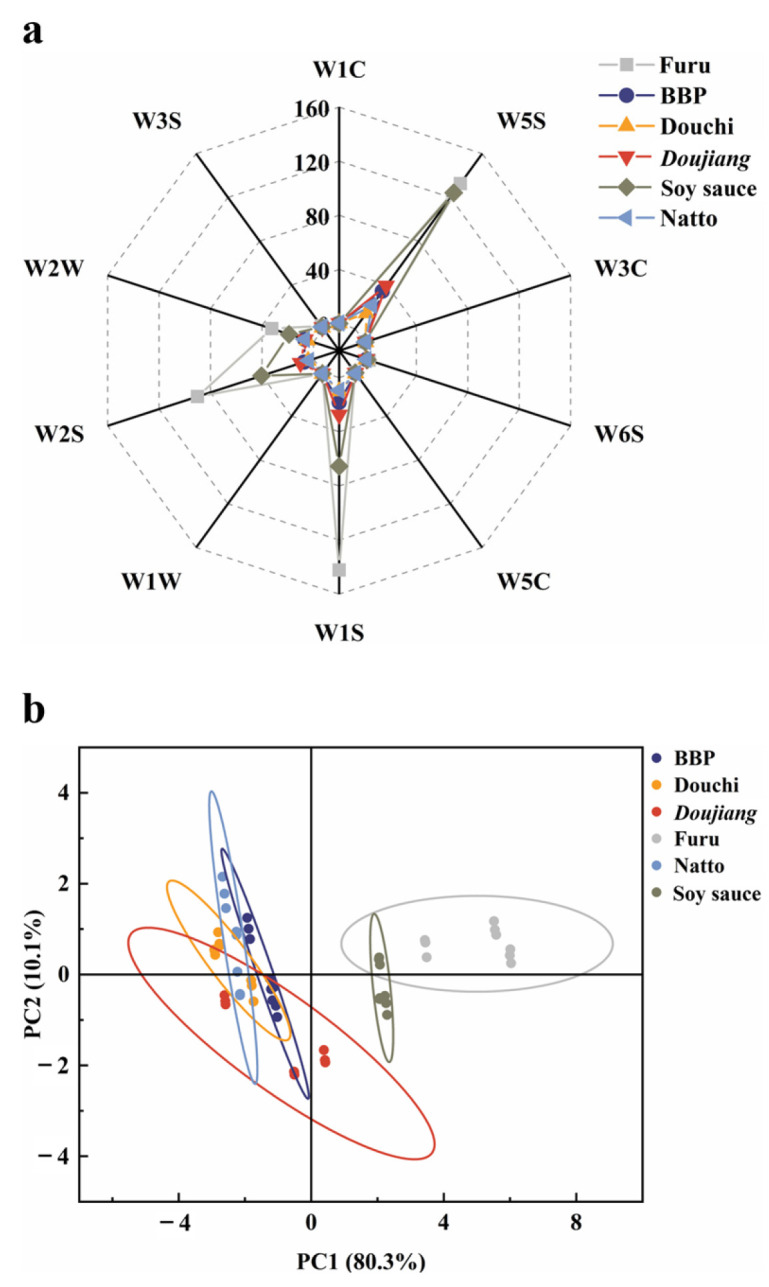
The volatile compounds of different fermented soy foods based on E-nose data: (**a**) radar chart and (**b**) PCA scores.

**Figure 3 foods-13-00415-f003:**
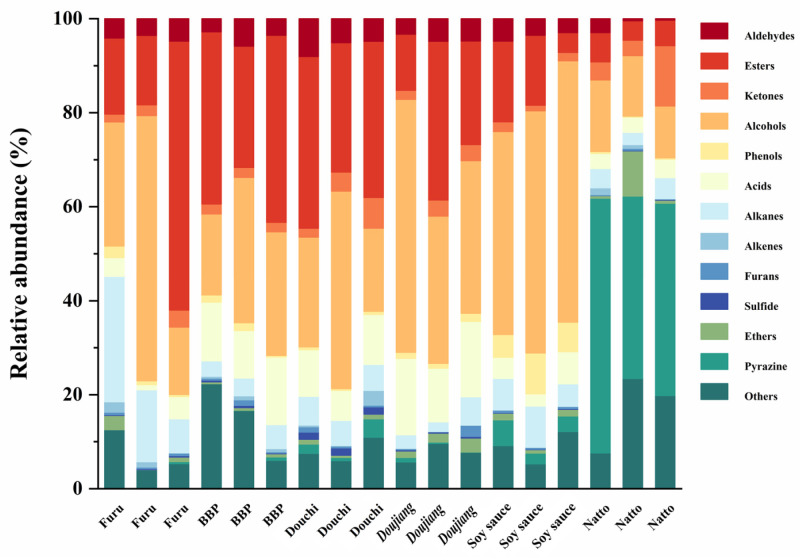
The volatile compounds in different fermented foods based on GC-MS data.

**Figure 4 foods-13-00415-f004:**
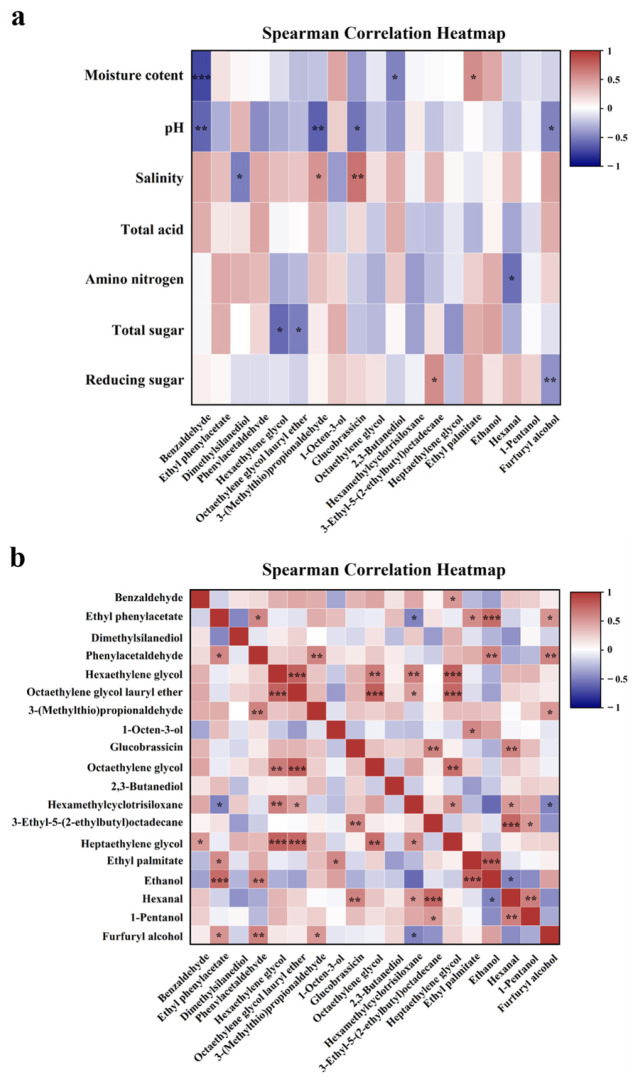
Heat map of the correlation between key physicochemical parameters and flavor compounds (**a**) and between flavor compounds (**b**) for six different fermented soy foods. Notes: * represents the correlation between physicochemical parameters (***: *p* < 0.001, **: *p* < 0.01, *: *p* < 0.05).

**Table 1 foods-13-00415-t001:** Statistics on the sources of fermented soy foods used in the study.

Sample Type	Brand	Ingredients
Furu	Liubiju	Water, Soybeans, Edible salt, Edible alcohol, White sugar, Wheat flour, Spices, Food additives
	Wangzhihe	Water, Soybeans, Edible alcohol, Edible salt, White sugar, Wheat flour, Red yeast rice, Magnesium chloride, Spices
	Guanghe	Water, Soybeans, Edible salt, Edible alcohol, White sugar, Wheat flour, Red yeast rice, Monosodium glutamate, Disodium 5′-presentation nucleotide, Sucralose
Broad bean paste (BBP)	Wanghong	Red chili peppers, Fava bean kernels, Edible salt, Wheat flour, Vegetable oil, Potassium sorbate
	Juancheng	Red chili peppers, Fava beans, Edible salt, Wheat flour, Canola oil, Spices, Potassium sorbate
	Dandan	Red chili peppers, Fava bean kernels, Edible salt, Water, Wheat flour, Canola oil, Potassium sorbate
Douchi	Waizumu	Soybeans, Water, Edible salt
	Chuannan	Soybeans, Water, Edible salt, Rice wine, Baijiu, Spices
	Hexian	Black beans, Water, Edible salt, Soy sauce, Brown sugar, Ginger, White sugar, Chi-flavor Baijiu, Yeast extract, Spices
*Doujiang*	Xinhe	Water, Soybean, Wheat flour, Edible salt, White sugar, Monosodium glutamate, Xanthan gum, Disodium 5′-inosinate, Potassium sorbate, Sucralose, Monopotassium glycyrrhizinate
	Haitian	Water, Soybeans, White sugar, Edible salt, Wheat flour, Monosodium glutamate, Vinegar, Disodium 5′-presentation nucleotide, Xanthan gum, Potassium sorbate, Sucralose
	Puning	Soybean, Edible salt, Wheat flour, Water, Sodium benzoate, Acesulfame, Monosodium glutamate
Soy sauce	Haitian	Water, Soybeans, Wheat, Edible salt, White sugar
	Xinhe	Water, Defatted soybeans, Wheat, Edible Salt, White sugar
	Wanzhuang	Water, Black beans, Defatted soybeans, Wheat, Edible salt, Fructose syrup, Monosodium glutamate, White sugar, Yeast extract, Disodium nucleotides, Sodium citrate, Ammonium glycyrrhizinate
Natto	Yamadai	Soybeans, Water, *Bacillus natto*
	Binari	Soybeans, Water, *Bacillus natto*
	Yanjing	Soybeans, Water, *Bacillus natto*

**Table 2 foods-13-00415-t002:** The sensitivities of sensors contained in the PEN3 electronic nose sensor array.

Sensor Number	Sensor Name	Performance Description
1	W1C	Aromatic compounds
2	W5S	Broad range, react on nitrogen oxides
3	W3C	Ammonia, aromatic compounds
4	W6S	Hydrocarbons
5	W5C	Alkanes, aromatic compounds
6	W1S	Methane, broad range of compounds
7	W1W	Sulfur compounds, terpenes
8	W2S	Broad range, alcohols
9	W2W	Organic sulfur compounds
10	W3S	Methane, aliphatic compounds

**Table 3 foods-13-00415-t003:** Physicochemical properties of different fermented soy foods.

Samples	Moisture Content (%)	pH	Salinity (g/100 g)	Total Acid (g/100 g)	Amino Nitrogen (g/100 g)	Reducing Sugar (g/100 g)
Furu-1	64.7 ± 0.6 ^abcde^	7.4 ± 0.0 ^c^	7.6 ± 0.3 ^i^	0.5 ± 0.0 ^j^	0.4 ± 0.0 ^j^	6.8 ± 0.0 ^b^
Furu-2	70.2 ± 0.4 ^a^	7.1 ± 0.0 ^d^	7.3 ± 0.4 ^i^	0.6 ± 0.0 ^j^	0.5 ± 0.0 ^i^	7.4 ± 0.0 ^a^
Furu-3	67.9 ± 4.0 ^ab^	6.7 ± 0.0 ^e^	8.9 ± 0.4 ^h^	0.8 ± 0.0 ^i^	0.4 ± 0.0 ^k^	6.5 ± 0.0 ^d^
BBP-1	49.4 ± 3.6 ^hi^	5.8 ± 0.0 ^ih^	16.1 ± 0.8 ^b^	0.9 ± 0.0 ^h^	0.3 ± 0.0 ^m^	3.7 ± 0.0 ^g^
BBP-2	42.4 ± 0.5 ^jk^	5.7 ± 0.0 ^i^	17.9 ± 0.4 ^a^	0.8 ± 0.0 ^i^	0.2 ± 0.0 ^p^	2.7 ± 0.0 ^k^
BBP-3	41.4 ± 0.4 ^kl^	5.4 ± 0.0 ^j^	16.7 ± 0.2 ^b^	1.2 ± 0.0 ^f^	0.3 ± 0.0 ^o^	3.3 ± 0.0 ^h^
Douchi-1	36.3 ± 0.3 ^l^	5.9 ± 0.0 ^gh^	10.7 ± 0.3 ^ef^	1.8 ± 0.0 ^b^	1.2 ± 0.0 ^b^	6.1 ± 0.0 ^e^
Douchi-2	47.3 ± 0.4 ^ij^	5.7 ± 0.0 ^i^	11.5 ± 0.2 ^cd^	1.8 ± 0.0 ^b^	0.4 ± 0.0 ^l^	5.0 ± 0.0 ^f^
Douchi-3	21.6 ± 0.5 ^m^	5.4 ± 0.0 ^j^	12.3 ± 0.2 ^c^	2.2 ± 0.1 ^a^	1.1 ± 0.0 ^d^	6.9 ± 0.0 ^b^
*Doujiang*-1	65.5 ± 0.5 ^abcd^	5.9 ± 0.0 ^g^	9.2 ± 0.3 ^gh^	0.8 ± 0.0 ^i^	1.1 ± 0.0 ^c^	2.6 ± 0.0 ^l^
*Doujiang*-2	54.9 ± 0.4 ^gh^	5.9 ± 0.0 ^gh^	9.9 ± 0.3 ^fg^	1.1 ± 0.0 ^fg^	0.7 ± 0.0 ^g^	6.6 ± 0.0 ^c^
*Doujiang*-3	65.6 ± 0.3 ^abc^	6.3 ± 0.1 ^f^	10.9 ± 0.3 ^de^	0.7 ± 0.0 ^j^	0.3 ± 0.0 ^m^	2.4 ± 0.0 ^m^
Soy sauce-1	59.9 ± 0.5 ^defg^	5.9 ± 0.0 ^g^	10.4 ± 0.3 ^ef^	1.4 ± 0.0 ^d^	0.9 ± 0.0 ^e^	1.4 ± 0.0 ^o^
Soy sauce-2	61.4 ± 0.4 ^cdef^	5.8 ± 0.0 ^hi^	11.8 ± 0.1 ^c^	1.6 ± 0.0 ^c^	0.9 ± 0.0 ^e^	3.2 ± 0.0 ^i^
Soy sauce-3	63.9 ± 0.6 ^bcde^	6.2 ± 0.03	11.7 ± 0.3 ^cd^	1.3 ± 0.0 ^e^	1.3 ± 0.0 ^a^	0.9 ± 0.0 ^q^
Natto-1	56.0 ± 0.4 ^fg^	7.5 ± 0.1 ^b^	0.4 ± 0.1 ^j^	0.9 ± 0.0 ^i^	0.7 ± 0.0 ^f^	1.1 ± 0.0 ^p^
Natto-2	52.3 ± 0.6 ^efg^	7.6 ± 0.1 ^b^	0.2 ± 0.0 ^j^	1.1 ± 0.0 ^g^	0.3 ± 0.0 ^m^	2.3 ± 0.0 ^n^
Natto-3	64.4 ± 0.4 ^bcde^	7.9 ± 0.1 ^a^	0.5 ± 0.0 ^j^	0.9 ± 0.0 ^h^	0.5 ± 0.0 ^h^	2.8 ± 0.0 ^j^

Physicochemical properties are represented as the mean ± SD obtained across triplicate measurements. Means with different superscript letters are significantly different horizontally (*p* < 0.05).

**Table 4 foods-13-00415-t004:** ROAV analysis of aroma compounds in different fermented soy foods.

Key Compounds	Odor Threshold (μg/kg)	ROAV	Odor Description
Furu			
Ethyl caproate	1	100.000	Apple peel, fruit
Tetradecane	1.1	47.651	Alkane
Ethyl caprylate	5	15.663	Fruit, fat
Ethyl isobutyrate	0.2	6.820	Sweet, ethereal fruity, alcoholic, fusel, rummy
Ethyl butanoate	1	2.986	Fruity, juicy, fruit pineapple, cognac
Ethyl caprate	1	1.223	Grape
Ethyl heptanoate	2	1.115	Fruit
Phenylacetaldehyde	4	1.073	Green, sweet, floral, hyacinth, clover, honey, cocoa
BBP			
Ethyl 2-methylbutanoate	0.2	100.000	Musty, potato, tomato, earthy, vegetable, creamy
Ethyl butanoate	1	53.193	Fruity, juicy, fruit pineapple, cognac
Methyl salicylate	0.06	14.429	Wintergreen, mint
Ethyl caproate	1	12.645	Sweet, fruity, pineapple, waxy, green, banana
Isoamyl acetate1	1.6	11.934	Sweet, fruity, banana, solvent
Hexanal	4.5	3.152	Fresh, green, fatty, aldehydic grass, leafy, fruity sweaty
alpha-Terpineol	0.3	2.595	Oil, anise, mint
Butyric acid	3.19	1.526	Rancid, cheese, sweat
Phenylacetaldehyde	4	1.116	Green, sweet, floral, hyacinth, clover, honey, cocoa
1-Octen-3-ol	1.5	1.017	Mushroom, earthy, green, oily, fungal, raw chicken
Douchi			
3-(Methylthio)propionaldehyde	2	100.000	Musty, potato, tomato, earthy, vegetable, creamy
Dimethyl disulfide	1.1	64.261	Onion, cabbage, putrid
Phenylacetaldehyde	4	48.130	Green, sweet, floral, hyacinth, clover, honey, cocoa
1-Octen-3-ol	1.5	44.682	Mushroom, earthy, green, oily, fungal, raw chicken
2,6-Dimethylpyrazine	10	28.824	Roasted nut, cocoa, roast beef
Dimethyl trisulfide	2.5	12.763	Sulfur, fish, cabbage
Hexanal	4.5	6.751	Fresh, green, fatty, aldehydic grass, leafy, fruity, sweaty
2,3-Butanediol	95.1	4.039	Fruity, creamy, buttery
Ethyl phenylacetate	1	2.443	Sweet, floral, honey, rose, balsam, cocoa
Benzaldehyde	350	1.088	Strong, sharp, sweet, bitter, almond, cherry
*Doujiang*			
3-(Methylthio)propionaldehyde	2	100.000	Musty, potato, tomato, earthy, vegetable, creamy
1-Octen-3-ol	1.5	76.316	Mushroom, earthy, green, oily, fungal, raw chicken
Phenylacetaldehyde	4	69.737	Green, sweet, floral, hyacinth, clover, honey, cocoa
Ethyl caprylate	5	23.684	Fruity, wine, waxy, sweet, apricot, banana, brandy, pear
Ethyl phenylacetate	1	22.368	Sweet, floral, honey, rose, balsam, cocoa
Dimethyl disulfide	1.1	18.740	Onion, cabbage, putrid
2,3-Butanediol	95.1	3.461	Fruity, creamy, buttery
Ethyl benzoate	60	1.308	Fruity, dry, musty, sweet, wintergreen
4-Hydroxy-2,5-dimethyl-3(2H) furanone	25	1.035	Sweet, cotton, candy, caramel, strawberry, sugar
Soy sauce			
2,3,5-Trimethylpyrazine	0.4	100.000	Roast, potato, must
2-Ethyl-5-methylpyrazine	0.4	88.431	Fruit, sweet
Phenylacetaldehyde	4	35.547	Green, sweet, floral, hyacinth, clover, honey, cocoa
2-Isobutyl-3-methylpyrazine	0.5	19.778	-
1-Octen-3-ol	1.5	18.764	Mushroom, earthy, green, oily, fungal, raw, chicken
Ethyl phenylacetate	1	10.460	Sweet, floral, honey, rose, balsam, cocoa
3-(Methylthio)propionaldehyde	2	9.382	Musty, potato, tomato, earthy, vegetable, creamy
2-Ethyl-3,5-dimethylpyrazine	2.2	9.365	Potato, roast
2,6-Dimethylpyrazine	10	8.830	Roasted nut, cocoa, roast beef
4-Ethyl-2-methoxyphenol	50	3.989	Spicy, smoky, bacon, phenolic, clove
2,3-Butanediol	95.1	1.699	Fruity, creamy, buttery
3-Methyl-1-butanol	220	1.134	Fusel, oil, alcoholic, whiskey, fruity, banana
Natto			
2,3,5-Trimethylpyrazine	0.4	100.000	Roast, potato, must
2-Ethyl-3,5-dimethylpyrazine	2.2	1.435	Potato, roast
2-Ethyl-5-methylpyrazine	0.4	1.173	Fruit, sweet

Odor threshold values were obtained from a book titled “Compilations of odor threshold values in air, water and other media” (edition 2011). Odor descriptions were obtained from the website http://www.thegoodscentscompany.com/ (accessed on 6 June 2023) and http://www.flavornet.org/ (accessed on 6 June 2023).

**Table 5 foods-13-00415-t005:** In vitro antioxidant activity of different fermented soy foods.

Samples	DPPH Radical Scavenging Activity (%)	ABTS^+^ Radical Scavenging Activity (%)	Hydroxyl Radical Scavenging Activity (%)	Superoxide Anion Radical Scavenging Activity (%)
Furu-1	59.3 ± 2.3 ^b^	1.9 ± 0.4 ^gh^	27.2 ± 0.3 ^fghi^	70.0 ± 2.7 ^e^
Furu-2	49.5 ± 7.9 ^bc^	2.3 ± 0.9 ^fgh^	39.9 ± 1.6 ^c^	79.4 ± 3.5 ^bcd^
Furu-3	26.3 ± 6.7 ^d^	1.4 ± 0.6 ^gh^	26.1 ± 1.4 ^ghi^	74.4 ± 4.9 ^de^
BBP-1	32.5 ± 3.1 ^d^	3.8 ± 0.4 ^defg^	33.8 ± 1.0 ^d^	90.9 ± 1.7 ^a^
BBP-2	53.7 ± 3.9 ^b^	0.8 ± 0.1 ^h^	31.3 ± 0.5 ^de^	77.1 ± 2.5 ^cde^
BBP-3	31.2 ± 3.6 ^d^	2.9 ± 0.9 ^efgh^	39.3 ± 1.0 ^c^	72.0 ± 4.8 ^de^
Douchi-1	49.3 ± 2.6 ^bc^	7.6 ± 1.1 ^b^	40.1 ± 1.8 ^c^	28.8 ± 0.9 ^f^
Douchi-2	57.3 ± 0.4 ^b^	12.3 ± 0.9 ^a^	45.9 ± 1.5 ^b^	26.2 ± 6.5 ^f^
Douchi-3	58.3 ± 3.4 ^b^	7.8 ± 1.3 ^b^	52.1 ± 0.1 ^a^	27.6 ± 4.6 ^f^
*Doujiang*-1	54.4 ± 12.5 ^b^	3.7 ± 0.5 ^defg^	30.5 ± 1.3 ^def^	90.9 ± 1.7 ^a^
*Doujiang*-2	34.4 ± 2.4 ^cd^	5.9 ± 1.0 ^bcd^	33.3 ± 0.6 ^d^	77.1 ± 2.5 ^cde^
*Doujiang*-3	28.9 ± 10.2 ^d^	6.5 ± 1.6 ^bc^	24.9 ± 1.2 ^hi^	72.0 ± 4.8 ^de^
Soy sauce-1	91.6 ± 6.1 ^a^	4.5 ± 0.5 ^cdef^	32.4 ± 0.9 ^de^	25.1 ± 2.0 ^f^
Soy sauce-2	88.3 ± 9.4 ^a^	5.0 ± 1.1 ^de^	28.5 ± 1.0 ^efgh^	22.5 ± 0.7 ^f^
Soy sauce-3	80.1 ± 3.9 ^a^	6.4 ± 1.5 ^bc^	31.8 ± 2.5 ^de^	25.0 ± 4.4 ^f^
Natto-1	53.8 ± 2.6 ^b^	12.36 ± 1.4 ^a^	24.4 ± 1.0 ^i^	94.6 ± 2.1 ^a^
Natto-2	52.0 ± 11.5 ^b^	13.8 ± 1.1 ^a^	25.8 ± 3.7 ^ghi^	86.0 ± 6.1 ^abc^
Natto-3	35.2 ± 4.7 ^cd^	13.9 ± 0.5 ^a^	29.0 ± 1.9 ^efg^	86.3 ± 0.7 ^ab^

Physicochemical properties are represented as the mean ± SD obtained across triplicate measurements. Means with different superscript letters are significantly different horizontally (*p* < 0.05).

**Table 6 foods-13-00415-t006:** In vitro antimicrobial activity of different fermented soy foods.

Samples	*Escherichia coli*	*Staphylococcus aureus*	*Listeria monocytogenes*	*Listeria seeligeri*
Furu-1	38.8 ± 4.8 ^de^	36.7 ± 7.4 ^cdef^	64.2 ± 2.1 ^bcde^	43.9 ± 1.7 ^def^
Furu-2	37.2 ± 2.0 ^def^	30.5 ± 2.4 ^fgh^	50.0 ± 3.7 ^ghi^	32.5 ± 2.7 ^gh^
Furu-3	28.1 ± 6.7 ^fgh^	32.0 ± 2.7 ^efgh^	56.7 ± 5.9 ^efg^	29.3 ± 4.1 ^h^
BBP-1	60.3 ± 3.5 ^ab^	39.1 ± 1.0 ^cde^	73.3 ± 3.0 ^a^	57.7 ± 5.7 ^abc^
BBP-2	66.1 ± 3.7 ^a^	33.6 ± 2.6 ^defg^	70.0 ± 3.9 ^abc^	65.0 ± 7.3 ^a^
BBP-3	58.7 ± 6.7 ^a^b	37.5 ± 3.6 ^cdef^	66. 7 ± 5.2 ^abcd^	54.5 ± 4.0 ^abcd^
Douchi-1	45.5 ± 2.2 ^dc^	51.6 ± 3.1 ^ab^	64.2 ± 2.8 ^bcde^	50.4 ± 5.0 ^cdef^
Douchi-2	32.2 ± 5.1 ^efgh^	37.5 ± 2.4 ^cdef^	52.5 ± 1.8 ^fgh^	52.8 ± 2.3 ^bcde^
Douchi-3	52.1 ± 2.3 ^bc^	41.4 ± 2.8 ^cd^	61.7 ± 0.9 ^cde^	65.0 ± 2.0 ^a^
*Doujiang*-1	24.8 ± 2.4 ^hij^	54.7 ± 2.4 ^a^	47.5 ± 4.7 ^hi^	61.0 ± 2.5 ^abc^
*Doujiang*-2	16.5 ± 2.4 ^ij^	41.4 ± 2.7 ^cd^	35.0 ± 2.4 ^jk^	40.7 ± 4.5 ^fg^
*Doujiang*-3	15.7 ± 3.2 ^j^	44.5 ± 4.2 ^bc^	41.7 ± 1.7	43.1 ± 5.1 ^efg^
Soy sauce-1	37.2 ± 3.5 ^def^	14.8 ± 2.9 ^j^	70.0 ± 2.4 ^abc^	41.5 ± 5.5 ^fg^
Soy sauce-2	25.6 ± 2.7 ^hi^	21.1 ± 4.9 ^ij^	60.8 ± 3.5 ^def^	28.5 ± 1.6 ^h^
Soy sauce-3	31.4 ± 2.0 ^efgh^	25.0 ± 2.2 ^hi^	70.8 ± 4.5 ^ab^	43.9 ± 1.5 ^def^
Natto-1	32.2 ± 2.8 ^efgh^	31.3 ± 2.1 ^efgh^	29.2 ± 1.2 ^kl^	50.4 ± 4.2 ^cdef^
Natto-2	35.5 ± 5.5 ^fg^	25.8 ± 2.1 ^ghi^	38.3 ± 5.6 ^j^	57.7 ± 6.2 ^abc^
Natto-3	27.3 ± 2.7 ^gh^	27.3 ± 1.8 ^ghi^	21.7 ± 3.8 ^l^	61.8 ± 5.4 ^ab^

Physicochemical properties are represented as the mean ± SD obtained across triplicate measurements. Means with different superscript letters are significantly different horizontally (*p* < 0.05).

## Data Availability

Data are contained within the article and Appendix A.

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
