# Peer review of "A Systematic Comparative Study on the Physicochemical Properties, Volatile Compounds, and Biological Activity of Typical Fermented Soy Foods"

_foods, 2024, doi:10.3390/foods13030415_

Round 1
Reviewer 1 Report
Comments and Suggestions for Authors
Dear Authors,
After reading the paper, it is clear that the authors are very experience in the research area of volatile compounds, and biological activity of fermented soy products. The Authors performed a lot of experiments and analyzes in order to obtain detailed research results. See my comments below, which most of them should be helpful in a clarification to readers from other countries outside of Asia:
1. Lines 68-70 – The authors indicates, that fermented food products are dependent on soybean varieties used for fermentation. In this place you should write, that most of soybean (more than 95%) are genetically modified and number of varieties drastically decreases because of the seed marked is dominated by large seed and chemical companies.
2. Lines 103-104 – research concerning on estimation of 6 different fermented foods products based on soy bean. It needs more detailed explanation of each, especially for readers from European countries. You should explain what is Furu. It is fermented tofu, also called fermented bean curd or white bean-curd cheese, tofu cheese. In English it is sometimes referred to as "soy cheese". Similar, what is the broad bean paste? It is fermented bean paste typically foods made from ground soybeans, which are indigenous to the cuisines of East, South and Southeast Asia. On other hand, what is Douchi, also known as fermented black soybeans or Chinese fermented black beans. And what is Doujiang, that is served in a bowl, and dounai is served in a cup. Soy sauce is well known around the Word that you did not must explain, but Natto is another product, from Asian cuisine, however it is a traditional Japanese food made from whole soybeans that have been fermented with Bacillus subtilis var. natto.
3. Lines 116-117 – Formula 1 should be write in symbols, e.g.: Mc as Moisture content, m – mass of wet sample, md – dry mass of sample. That the formula should looks as follow:
see in file
Under formula you should listed whole description of each symbol.
Please change it similar in formula 2 (lines 183-184), and fulfill description of each symbol in equation in other formulas 3-6.
4. Lines 142-150 – If it possible, please compare E-nose technique to other publication describing gas sensors coupled with chemometrics methods or rapeseed quality degradation on the reaction of MOS type sensor-array.
5. Lines 293-294 – Description on the figure 1 is completely invisible; should be enlarged.
6. Lines 323-324 – The figure 2 and description on it is completely invisible, should be enlarged.
7. Lines 416-417 – Description on the figure 4 is completely invisible, should be enlarged.
E-nose analysis should be compared to other methods based on reaction of metal oxide semiconductor (MOS) e.g. “Influence of changes in the level of volatile compounds emitted during rapeseed quality degradation on the reaction of MOS type sensor-array” or “Non-destructive test to detect adulteration of rice using gas sensors coupled with chemometrics methods”.
In general, the manuscript is almost excellent; the manuscript is adequate to the results presented in this paper; the structure of the work is clear and complete. After reading the paper, it is clear that the authors have experience in fermented soy foods and the structure of presented work is clear and looks complete.

Nomenclature concerning fermented soy foods is proper, as well as description of methods used in this paper.
Author Response
Thank you very much for taking the time to review this manuscript. Those comments are all valuable and helpful for revising and improving our paper and the essential guiding significance of our research. We have studied the comments carefully and made corrections, which we hope meet with approval. Please find the detailed responses below and the corresponding corrections highlighted

Reviewer 2 Report
Comments and Suggestions for Authors
Dear Authors
A systematic comparison of different fermented soy foods was performed using E-nose, HS-SMPE-GC×GC-MS, bioactivity validation and correlation analysis. However, I can send only one observation to improve the manuscript:
Keywords. I recommend that the keywords are not repeated in the title.
Materials and Methods
Table 1. Place scientific name of bacteria in italics.
Pag. 3, line 105. Was homogenization manual or using some equipment? need to specify.
Pag. 3, line 106-107. I recommend adding equipment data (brand, model, place) in the centrifugation and ultrasonication step.
Pag. 4. I recommend you add reference to some physicochemical techniques that you mention in methodology 2.3.
Pag. 5, line 162. Delete a parenthesis.
Pag. 5, line 177. I recommend updating reference.
Pag. 5, line 182, 193, 204, 214. It is important to place the equipment data where the absorbance was measured.
Results and discussion
Table 3, 5 and 6. To have a cleaner table, I recommend leaving your results one tenth after the point.
I recommend improving the sharpness of figure 1 and 4.
Author Response
Thank you very much for taking the time to review this manuscript. Those comments are valuable and helpful for revising and improving our paper and the essential guiding significance to our research. We have studied the comments carefully and made corrections, which we hope meet with approval. Please find the detailed responses below and the corresponding corrections highlighted.

Reviewer 3 Report
Comments and Suggestions for Authors
Introduction
Please provide more information regarding the research gap. This could include a brief summary of what is already known about the FSFs in question and gaps that your study aims to fill.
Materials and methods
Line 100: The selection of FSFs and brands is well described. However, it would be beneficial to include a rationale for the choice of these specific brands and types of FSFs. Is there a particular reason these brands were chosen, such as market share or geographic representation?
Line 104: The sample preparation procedure is detailed, which is good. However, it would be helpful to explain why these specific conditions (e.g., 50 mL of distilled water, 30 min ultrasonic extraction at 35°C) were chosen. Are these standard conditions for such analyses, or were they determined experimentally?
Results and discussion
L236: The analysis of moisture content, pH, salinity, total acid, amino nitrogen, and reducing sugars across different FSFs is well presented but it would be beneficial to discuss how these properties directly relate to consumer preferences or health benefits.
L282: The correlation analysis between different physicochemical properties (Fig. 1) is insightful. How would like this section to be expanded: please delve deeper into why these correlations exist and their implications for the fermentation process and final product quality.
L300: Please explain how the response of the E-nose sensors translates to human sensory perception would be helpful.
L332: Linking the flavor compounds with specific fermentation processes or microbial actions can provide a deeper understanding of the fermentation science involved.
L424: It would be useful to compare these activities with those found in non-fermented soy products to highlight the benefits of fermentation.
The discussion could be expanded to include potential applications of the findings, comparisons with other studies, and implications for future research.
Ensure consistency in terminology and measurement units throughout the manuscript.
Briefly acknowledge any limitations of your study, such as the selection of FSFs
Comments on the Quality of English LanguageThe manuscript would benefit from moderate editing to improve the English language, sentence structure, and grammatical accuracy. While the scientific content is clear, refining the language will enhance readability and professional presentation
Author Response

(The authors gave the same response as above.)
